# Direct Evidence for Viral Antigen Presentation during Latent Cytomegalovirus Infection

**DOI:** 10.3390/pathogens10060731

**Published:** 2021-06-10

**Authors:** Niels A. W. Lemmermann, Matthias J. Reddehase

**Affiliations:** Institute for Virology and Research Center for Immunotherapy (FZI), University Medical Center of the Johannes Gutenberg-University Mainz, 55131 Mainz, Germany; matthias.reddehase@uni-mainz.de

**Keywords:** antigen presentation, cytomegalovirus, inflationary effector-memory CD8 T cells (iTEM), latent infection, memory inflation (MI), viral latency

## Abstract

Murine models of cytomegalovirus (CMV) infection have revealed an immunological phenomenon known as “memory inflation” (MI). After a peak of a primary CD8^+^ T-cell response, the pool of epitope-specific cells contracts in parallel to the resolution of productive infection and the establishment of a latent infection, referred to as “latency.” CMV latency is associated with an increase in the number of cells specific for certain viral epitopes over time. The inflationary subset was identified as effector-memory T cells (iTEM) characterized by the cell surface phenotype KLRG1^+^CD127^−^CD62L^−^. As we have shown recently, latent viral genomes are not transcriptionally silent. Rather, viral genes are sporadically desilenced in a stochastic fashion. The current hypothesis proposes MI to be driven by presented viral antigenic peptides encoded by the corresponding, stochastically expressed viral genes. Although this mechanism suggests itself, independent evidence for antigen presentation during viral latency is pending. Here we fill this gap by showing that T cell-receptor transgenic OT-I cells that are specific for peptide SIINFEKL proliferate upon adoptive cell transfer in C57BL/6 recipients latently infected with murine CMV encoding SIINFEKL (mCMV-SIINFEKL), but not in those latently infected with mCMV-SIINFEKA, in which antigenicity is lost by mutation L8A of the C-terminal amino acid residue.

## 1. Introduction

Murine models of cytomegalovirus (CMV) infection have revealed unusual kinetics of the CD8^+^ T-cell response to systemic infection, hallmarked by an expanding pool size of cells specific for certain viral epitopes over time. This phenomenon became popular under the catchphrase “memory inflation” (MI) and was studied extensively (for reviews, see [1,2,3,4,5,6]). MI raised interest because of the promising perspective of using CMVs as vaccine vectors by replacing MI-driving intrinsic antigenic peptides with peptides of unrelated pathogens or tumors to induce a self-perpetuating and even enhancing memory specific for the vaccine target [7,8,9,10,11,12]. As with conventional protein antigens and in non-chronic viral infections, the peak of the primary CD8^+^ T-cell response to presented CMV epitopes is first followed by pool contraction and establishment of immunological memory capable of mounting a recall response. This primary response resolves the pathogenic acute infection and thus protects against organ disease, but fails to clear the viral genomes. Instead, the immunological contraction phase is paralleled by the establishment of a latent infection, also referred to as “latency.” Latency is defined as a state of non-productive infection, characterized by maintenance of the viral genome in certain cell types in a largely, though not completely, silenced state during which infectious virus is not produced [13,14]. Viral latency has nevertheless an impact on the CD8^+^ T-cell response in that it provides the molecular basis for MI that is characterized by a more or less steady increase in cell numbers over time [15].

The inflationary subset of functional, IFNγ-secreting and antivirally protective CD8^+^ T cells derived from pulmonary infiltrates of latently infected mice was originally phenotyped as being CD62L^−^ T effector-memory cells (TEM) [16,17], and with the description of the marker molecule KLRG1 [18] re-defined as short-lived effector cells (SLEC), characterized by the cell surface marker profile KLRG1^+^CD127^−^CD62L^−^ [19]. A more recent study showed an extended life span of lung-resident inflationary cells, as compared to acute response effector cells, based on IL15-mediated expression of the anti-apoptotic protein Bcl-2, which makes them “memory cell-like” [20]. We therefore proposed to re-name these cells “inflationary T effector-memory cells” (iTEM), distinguishing them from non-inflationary, conventional KLRG1^−^CD127^+^CD62L^−^ T effector-memory cells (cTEM) [21].

A requirement for frequent antigen encounters for the expansion of iTEM is implied already by the early finding that the expression of the iTEM-defining lead marker KLRG1 requires persistent or at least repetitive antigen stimulation. Specifically, KLRG1 was found to be expressed by CD8^+^ T cells during persistent infections but lost in resolved infections [18]. In support of MI being driven by antigen, a recent mathematical model proposed frequent restimulations by antigen to give optimal fit between model predictions and experimental data by leveling oscillations between iTEM pool contractions and expansions in the absence and presence of antigen, respectively [22].

At first glance, a requirement for antigen in the expansion and maintenance of the iTEM pool appears to be in conflict with the notion that iTEM are maintained in pulmonary infiltrates in a seemingly antigen-independent manner by IL15 derived from non-hematopoietic lung cells [20]. This conclusion, however, rested on the widely held assumption that direct antigen presentation as well as antigen cross-presentation both require ongoing productive infection and involve professional antigen-presenting cells (APC) of the hematopoietic lineage. During latent infection of mice with murine cytomegalovirus (mCMV), however, MI is driven by direct antigen presentation in latently infected non-hematopoietic cells [23,24], such as endothelial cells of the lung vasculature [25,26]. Intriguingly, probably the same cells also produce the IL15 that extends the life span of iTEM [20]. In line with this, differential high-density microarray analyses of cellular gene expression induced by the β-herpesvirus mCMV revealed an upregulation of the expression of both, IL15 and its receptor IL15R, which was not the case for the α-herpesviruses herpes simplex virus type-1 and type-2 [27].

As we have shown recently [26], latent viral genomes are not transcriptionally silent. Rather, viral immediate-early (IE), early (E), and late (L) genes [28,29,30] are sporadically desilenced in a stochastic fashion not following the coordinated, progressing cascade of IE-E-L gene expression of the viral productive program [26,31,32,33]. The current hypothesis proposes MI to be driven by presented viral antigenic peptides encoded by the respective, stochastically transcribed viral genes [15,26]. Notably, this stochastic gene expression and consequent sporadic presentation of MI-driving antigenic peptides give a molecular explanation for the immunological finding of stochastic expansions of iTEM clones in the course of MI [34].

Although this mechanism is most plausible, independent evidence for antigen presentation during viral latency is pending. Here we fill this gap by showing that T-cell-receptor (TCR) transgenic OT-I cells [35] that are specific for peptide SIINFEKL, which is presented by the MHC class-I molecule K^b^ [36], proliferate upon cell transfer in C57BL/6 recipients latently infected with recombinant mCMV engineered to encode SIINFEKL as a foreign antigenic peptide (mCMV-SIINFEKL). In contrast, OT-I cells fail to proliferate in recipients latently infected with virus mCMV-SIINFEKA, in which antigenicity of the epitope is lost by point mutation L8A of the C-terminal amino acid residue [37]. Importantly, as known intrinsic MI-driving epitopes presented in the *H-2^b^* haplotype, namely M38 and m139 [38,39], lead to MI after infection with either of the two viruses, the failure of OT-I proliferation in mice latently infected with mCMV-SIINFEKA is not caused by a putatively non-supportive cytokine milieu, as it likely is the case in CMV-naïve, uninfected transfer recipients, but can now definitively be attributed to the missing cognate antigen.

## 2. Materials and Methods

### 2.1. Mice, Viruses, and Establishment of Latent Infection

Female C57BL/6 (8-week-old, haplotype *H-2^b^*) mice were purchased from Harlan Laboratories and were housed under specified pathogen-free (SPF) conditions in the Translational Animal Research Center (TARC) of the University Medical Center of the Johannes Gutenberg-University Mainz. TCR-transgenic OT-I mice [35] were bred and housed in the TARC under SPF conditions.

Recombinant viruses mCMV-SIINFEKL and mCMV-SIINFEKA were generated by two-step replacement BAC mutagenesis, reconstitution, removal of BAC sequences, and purification, essentially as described previously ([37], and references therein). A sequence that codes for an endogenous D^d^-presented and MI-driving antigenic peptide in the non-essential gene *m164* was replaced with sequences coding for peptides SIINFEKL or SIINFEKA. For the sake of brevity, specific features included identically into both viruses, such as deletion of gene *m157* that would encode an activatory ligand for the Ly49^+^ natural killer cell subset, as well as integration of the gene encoding luciferase for other purposes, are not specified in the virus nomenclature.

Infection was performed with 10^6^ plaque-forming units of the respective viruses by intravenous administration. The establishment of latency in the definition by Roizman and Sears [13], namely maintenance of viral genome in the absence of infectious virus, was routinely controlled as documented previously [26].

### 2.2. Peptides and Quantitation of Functional Epitope-Specific CD8^+^ T Cells

Synthetic peptide SIINFEKL, which is presented by the MHC class-I molecule K^b^, and the also K^b^-presented mCMV peptides SSPPMFRV and TVYGFCLL, derived from mCMV open reading frames M38 and m139, respectively [39], were purchased from JPT Peptide Technologies (Berlin, Germany). Peptides were exogenously loaded on EL-4 (*H-2^b^*) lymphoma cells at a saturating concentration of 10^−7^ M for use as stimulator cells in an IFNγ-based enzyme-linked immunospot (ELISpot) assay. At indicated times after infection, CD8^+^ T cells were immunomagnetically purified from spleen cell suspensions of latently infected C57BL/6 mice (pool of 5 mice per time) to serve as responder cells. The ELISpot assay was used to quantitate functional, IFNγ-secreting, epitope-specific cells within the CD8^+^ T-cell population ([40], and references therein). In brief, graded numbers of CD8^+^ T cells were seeded with the peptide-loaded stimulator cells in triplicate cultures, and spots, each representing a specifically sensitized, IFNγ-secreting cell, were counted automatically, based on standardized criteria using ImmunoSpot S4 Pro Analyzer (Cellular Technology Limited, Cleveland, OH, USA). Frequencies and their 95% confidence intervals were determined by intercept-free linear regression analysis (Graph Pad Prism 6.04, Graph Pad Software, San Diego, CA, USA).

### 2.3. Cell Transfer and in Vivo Proliferation Assay

CD8^+^ T cells were immunomagnetically isolated from spleens of 10–20week-old OT-I mice. This yields an almost pure population of Vα2Vβ5 TCR-transgenic OT-I cells specific for the peptide-MHC class-I (pMHC-I) complex SIINFEKL-K^b^. OT-I cells were incubated for 4 min at 37 °C at a concentration of 1x10^7^ cells/mL with 5µM of 5(6)-carboxyfluoresceindiacetate (CFDA; Merck Darmstadt) in phosphate-buffered saline (PBS). CFDA converts intracellularly into the fluorescent dye carboxyfluorescein diacetate succinimidyl ester (CFSE). The reaction was stopped with FCS, and the cells were washed three times with PBS. 2 × 10^7^ CFSE-labeled OT-I cells were administered intravenously into groups of C57BL/6 mice (n = 5) that were either left uninfected or were latently infected for 15 wks with viruses mCMV-SIINFEKL or mCMV-SIINFEKA. CD8^+^ T cells were recovered at 60 h after OT-I cell transfer from pooled spleens as well as from pooled popliteal lymph nodes. Cell divisions of OT-I cells were determined by cytofluorometric analysis of the loss of CFSE fluorescence [41,42,43].

## 3. Results and Discussion

### 3.1. Epitope-Specific MI in C57BL/6 Mice Latently Infected with Recombinant mCMVs Expressing Antigenic Peptide SIINFEKL or its Non-Antigenic Analog SIINFEKA

For exploiting OT-I technology, we pursued the concept of using CMV, here mCMV, as a model of a vaccine vector in which an endogenous MI-driving antigenic peptide is genetically replaced with a foreign antigenic peptide of interest. The strategy is based on the idea that the foreign antigenic peptide should assume an MI phenotype when placed in the position of an established MI-driving endogenous antigenic peptide so that fundamental conditions for antigen presentation during viral latency are fulfilled. These include, in the first place, expression of the corresponding gene during latent infection [26], as well as a favorable localization within the carrier protein for efficient proteasomal processing [44,45,46], in particular by the constitutive proteasome [47], which is present in non-hematopoietic, latently infected tissue cells that are known to support MI [23,24].

In the specific case, we integrated the antigenic peptide SIINFEKL or its non-antigenic peptide analog SIINFEKA into the non-essential protein m164 in place of the known MI-driving, D^d^-presented antigenic peptide AGPPRYSRI ([21] and references therein). In latency, the peptides can be stochastically expressed from the authentic E-phase transcript, as well as from an IE-phase transcript initiating at an upstream promoter ([48,49], and references therein) (Figure 1A).

In C57BL/6 mice that were latently infected with mCMV-SIINFEKL, functional IFNγ^+^CD8^+^ T cells specific for SIINFEKL-K^b^ showed the typical kinetics of expansion, contraction, and almost steady inflation that defines MI (Figure 1B, top panel). Thus, consistent with the rationale, SIINFEKL indeed assumed an MI phenotype when integrated into a carrier protein in an MI-supporting context. Expectedly, no response to SIINFEKL was induced in mice latently infected with mCMV-SIINFEKA.

As a control, latent infection with either of these two viruses supported MI of IFNγ^+^CD8^+^ T cells specific for the known MI-driving, endogenous antigenic peptides M38 and m139 [38,39], both of which are also presented by K^b^ (Figure 1B, center and bottom panels, respectively). Logically, a point mutation that ablates antigenicity of a particular epitope is not expected to affect the response to unrelated other epitopes. Nevertheless, this control is important, as it proves that latent infection with mCMV-SIINFEKA also provides a cytokine milieu principally supportive of MI.

### 3.2. Antigen Presentation Is Essential for Driving OT-I Proliferation in Latently Infected Cell Transfer Recipients

Fluorescence-labeled OT-I served as reporter cells for detecting antigen presentation during latent mCMV infection. As an absolute baseline control, OT-I cells were transferred intravenously into mCMV-naïve, uninfected C57BL/6 recipient mice. The absence of proliferation in the spleen and the popliteal lymph node (PLN) excluded an antigen-independent proliferation that might have been triggered by cell stress associated with the transfer as such (Figure 2A). In contrast, transfer into recipients latently infected with virus mCMV-SIINFEKL led to OT-I proliferation at both lymphoid sites (Figure 2B). As OT-I cells that encounter antigen in latently infected peripheral tissues, such as the lungs ([26], and more references therein), are unlikely to home to a non-draining lymph node, OT-I proliferation in the PLN supports the previous conclusion that non-hematopoietic cells in lymph nodes are latently infected and can present antigen locally [24].

In the same report [24], proliferation data consistent with ours were obtained with a related strategy showing that lymphoid tissue-derived central memory T cells expressing a transgenic TCR specific for M38-K^b^ proliferated upon cell transfer in the lungs of latently infected but not of mCMV-naïve recipients. This suggested antigen presentation in latently infected lungs, in accordance with our previous reports on the stochastic expression of MI-driving viral genes during latency in the lungs ([15,26,49] and more references therein).

It is tempting to conclude already at this stage that the presence of the antigen makes the relevant difference between latently infected and CMV-naïve, uninfected recipients. However, besides the presence or absence of antigen, respectively, there is increasing evidence for a modulation of cellular gene expression by latent CMV infections. Importantly, latent hCMV infection has a profound effect on the cellular secretome, mediated not just by viral proteins expressed during latency but also through viral latency-associated changes in cellular microRNAs ([50,51], reviewed in [52,53]). Consistent with this, the already discussed finding that the prolonged life span of iTEM during mCMV latency, as compared to acute response effector cells, depends on IL15-mediated expression of the anti-apoptotic protein Bcl-2 [20] implies that IL15 is expressed not only after acute mCMV infection [27], but also during latent infection.

Accordingly, OT-I cell transfer into mCMV-naïve, uninfected recipients is not the final control for allowing us to conclude on antigen-dependence of OT-I proliferation, as this approach disregards the modulation of the cytokine milieu and possible unrecognized other microenvironmental remodeling associated with latent infection. In order to meet such objections, we performed the decisive experiment of transferring the OT-I cells into recipients latently infected with mCMV-SIINFEKA, which differs from mCMV-SIINFEKL just by a point mutation that selectively destroys the antigenicity of this particular epitope, while maintaining all features of latent infection, including a cytokine milieu supportive of MI specific for endogenous mCMV epitopes M38 and m139 (recall Figure 1B). This experiment gave the clear and ultimate answer that proliferation of OT-I cells upon transfer into latently infected recipients critically depends on the presentation of the cognate antigenic peptide (Figure 2C).

It is always a pertinent question if findings from any animal model system apply also to human infection in view of the obvious genetic differences in both the virus and the host genetics. Although virus-host co-evolution has led to biological convergence in many principles of pathogenesis and immune control of CMVs in their respective host species (discussed in [54]), the cellular sites of latency make a prominent difference, as we have reviewed recently by comparing mCMV and hCMV [14]. Specifically, whereas mCMV becomes latent in endothelial cells [15,26,55,56], prevailing cellular sites of hCMV latency are hematopoietic cell types of the myeloid differentiation lineage, which include progenitors of professional APC [57,58,59,60]. So, the statement that in mouse bone marrow chimeras MI is driven by non-hematopoietic cell types [23,24] is valid only in the mouse model and is there compatible with latency being established in EC, which represent a non-hematopoietic cell type. In these studies, a relevant contribution of hematopoietic lineage-derived, radiation-resistant APC of recipient-genotype in the chimeras was made unlikely by the finding that CD11c^+^ dendritic cells represent the cell type that is most completely replaced with donor-genotype [23,55]. Based on the view that a state of “dynamic latency,” characterized by sporadic expression of IE, E, and L genes not following the canonical temporal IE-E-L cascade of productive reactivation, exists in human hematopoietic progenitors of APC [31], we propose that such a stochastic gene expression, just as described for the murine latency model [15,26], is the driver of MI also in humans, unrelated to a potential APC function that anyway applies only to the mature end-stages of hematopoietic-lineage differentiation. Obviously, due to general constraints applying to human studies, this prediction will be difficult to verify by clinical investigation, especially since MI in humans is not consistently observed, depending on the individual and mostly unknown infection history [61,62].

As we have discussed in greater detail previously [14,26], it is difficult to distinguish between latent infection and low-level productive persistent infection and/or episodes of productive reactivation on the organismal level. Thus, a contribution of canonical IE-E-L gene expression to MI is not excluded formally, and there is no apodictic argument against a co-existence of both MI-driving ways of antigen presentation, except that stochastic viral gene expression can elegantly explain why immune evasion genes, which unavoidably become expressed during the lytic cycle, do not prevent MI [26]. As a contra-argument, one may propose that MI, at least in humans, is driven through antigen cross-presentation by uninfected APCs following persistent or reactivated productive infection, which in humans appear to occur more frequently than in the mouse models. In all this discussion, we find it important to consider that latent and persistent or reactivated infections are not mutually exclusive but can co-exist compartmentalized to different cell types and organs. For example, experimentally induced reactivation of latent mCMV in organs was found to be a stochastic process that can take place in any organ that harbors latent viral genomes, while at the same time other organs stay latently infected (reviewed in [62]). Similarly, during clinical hCMV latency, children can continue to shed low levels of virus from infected epithelial cells in the salivary glands or kidneys for months to years, while molecular latency is long established in hematopoietic myeloid lineage progenitor cells (for reviews, see [62,63,64]). As this shedding of infectious virus occurs via a secretory pathway directly into the salivary duct or renal tubules and collecting duct, respectively, it is questionable if the immune system is engaged at all. In fact, absence of immune sensing defines these sites as being “immunoprivileged”, and this may actually explain the persistence of virus shedding. We thus argue that persistent virus shedding is not a driver of MI in humans. Positively, we propose that, in both systems, stochastic expression of viral genes in the respective latently-infected cell types is the driver of MI.

## 4. Conclusions

Antigen presentation by latently infected, non-hematopoietic cells in host tissues has been inferred from indirect evidence, such as stochastic episodes of transcription from viral genes encoding MI-driving epitopes, as well as from maintenance of restimulation-dependent KLRG1 expression by the inflationary subset of CD8^+^ T cells, the iTEM. The proliferation of OT-I cells in cell transfer recipients that were latently infected with mCMV-SIINFEKL, expressing the cognate epitope, but not in those latently infected with the antigenicity loss mutant mCMV-SIINFEKA, shows that an MI-supporting milieu is not sufficient to drive OT-I proliferation. This result provides the first direct evidence for viral antigen presentation during mCMV latency.

## Figures and Tables

**Figure 1 pathogens-10-00731-f001:**
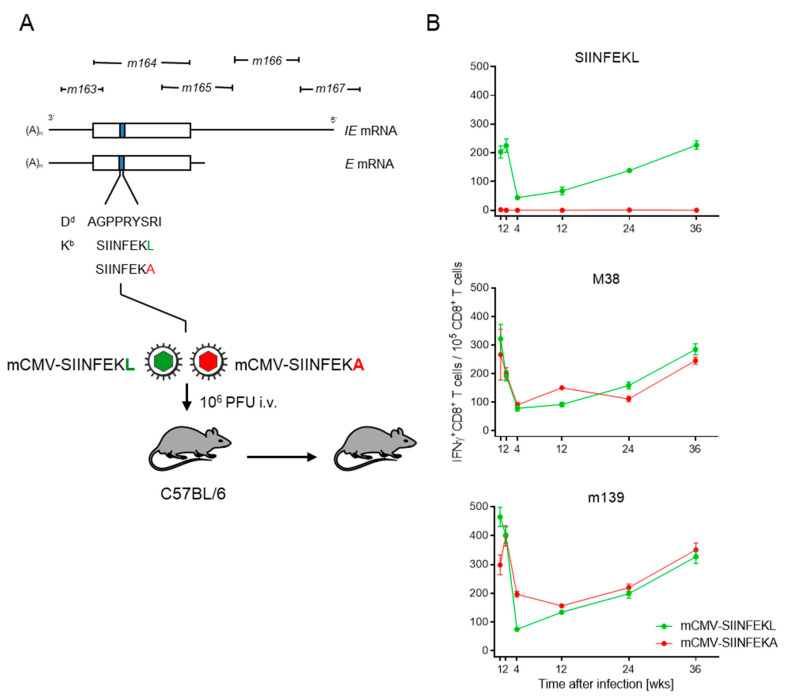
Verification of MI in latently infected C57BL/6 mice. (**A**) Map of the foreign peptide integration site for replacing the established MI-driving antigenic peptide m164 with peptides SIINFEKL and SIINFEKA, thus generating epitope-recombinant viruses mCMV-SIINFEKL and mCMV-SIINFEKA, respectively. IE, immediate early; E, early. C57BL/6 mice were infected intravenously (i.v) with 10^6^ plaque-forming units (PFU) of either of the two viruses. Empty boxes indicate the map position of the m164 open reading frame. (**B**) Kinetics of the IFNγ^+^CD8^+^ T-cell responses in mice latently infected with mCMV-SIINFEKL (green symbol) or mCMV-SIINFEKA (red symbol). Symbols represent numbers of responding cells and their 95% confidence limits, as determined by intercept-free linear regression analysis. (Top panel) target peptide SIINFEKL; (center panel) target peptide mCMV M38; (bottom panel) target peptide mCMV m139.

**Figure 2 pathogens-10-00731-f002:**
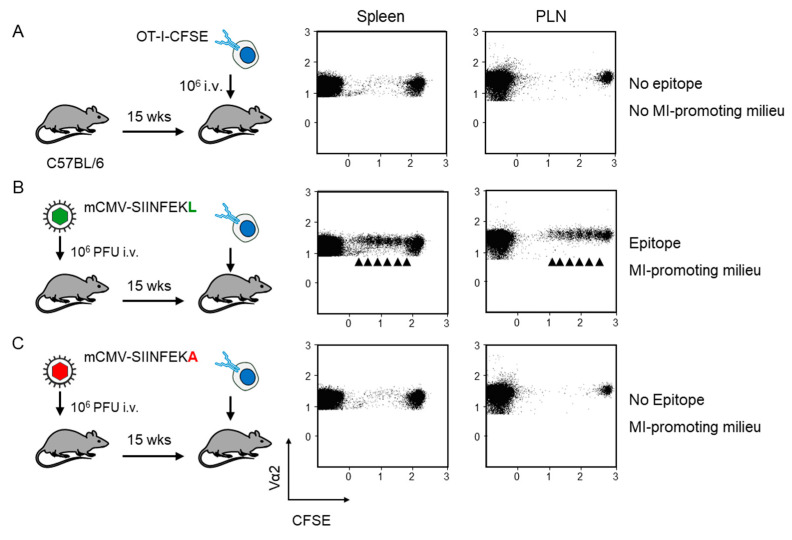
Verification of antigen presentation in latently infected C57BL/6 recipients of OT-I cell transfer. Fluorescence-labeled OT-I cells (2x10^7^ per mouse) were transferred intravenously (i.v) into C57Bl/6 mice as recipients. At 60 h after transfer, CD8^+^ T cells were recovered from the spleen and popliteal lymph nodes (PLN) of 5 mice per group, and proliferation of fluorescence-labeled TCR-Vα2^+^ OT-I cells was assessed by loss of the fluorescent marker with every cell division (arrows). Transfer recipients were: (**A**) Uninfected C57BL/6 mice; (**B**) C57BL/6 mice latently infected with mCMV-SIINFEKL, and (**C**) C57BL/6 mice latently infected with mCMV-SIINFEKA.

## Data Availability

The data presented in this study are available on request from the corresponding author.

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
