# Peer review of "Direct Evidence for Viral Antigen Presentation during Latent Cytomegalovirus Infection"

_pathogens, 2021, doi:10.3390/pathogens10060731_

Round 1
Reviewer 1 Report
The authors have answered a very important question concerning antigen presentation during the latency phase of murine cytomegalovirus infection using a well-characterized T cell epitope, SIINFEKL, inserted into an 'memory inflation gene', m164. The studies are well designed and straightforward and clearly demonstrate that antigen presentation occurs during latency. As m164 is a lytic gene expressed during the early phase of replication, this represents memory T cell recognition following reactivation of virus. Clearly for detection of m164 specific T cells by using specific tetramers, antigen presentation has to occur but by using a 'foreign epitope' and a mutant epitope, the authors provide a clever way to demonstrate that this is indeed occurring. For OT-I T cell proliferation to occur following adoptive transfer into latently infected mice, expression of the SIINFEKL epitope is necessary.
Minor revision: please add the days post infection into the methods.
Author Response
We thank the reviewer for his comment. We modified Methods and Fig 2 to make the experimental procedure more clear.
Reviewer 2 Report
This is an excellent manuscript in all regards, including logic, rationale, experimental approach, data presentation, data interpretation, and conclusions. This is a ‘simple but elegant’ demonstration that presentation of antigen by infected cells during latent MCMV infection drives memory inflation of antigen-specific T cells. This is a critical proof-of-concept emphasizing the dynamic virus-host relationship that occurs during long-term MCMV infection.
It might be useful to the reader to speculate on how these findings with MCMV relate to human CMV. Specifically, HCMV is similarly considered to enter into a state of latency in long-term infected hosts. Yet, natural history studies unequivocally demonstrate that virus can be shed in bodily fluids for long periods of time, or serially reactivate and be secreted into breast milk, for example. The question is whether there is some low level of HCMV gene expression in some chronically infected cells that is not stochastic (i.e., IE-E-L).
Author Response
We thank the reviewer for the valuable comments that help us to improve our manuscript. We enhanced the discussion by comparing differences in mCMV and hCMV latency. Please see lines 232-275.
Reviewer 3 Report
The brief report presented by Lemmermann and Reddehase focusses on the intersection of very topical areas of research in immunology and herpesvirology. Firstly, how tissue resident memory T cells are maintained and secondly, how memory inflation is driven in MCMV-infected mice long after acute infection has been resolved.
Using OT-1 T cells and mice infected with either SIINFEKL or a non-antigenic derivative SIINFEKA, the authors provide robust evidence for cognate antigen driving T cell proliferation in secondary lymphoid organs weeks after MCMV inoculation.
The remaining question concerns identifying the antigen presenting cells driving the iTEM reservoir. Here, the authors conclude that stochastic expression of latency-associated MCMV antigens, expressed by non-hematopoietic cells are the antigen-presenting targets. And it is a highly plausible conclusion: i.v. inoculation of MCMV would certainly infect a wide range of cellular targets – indeed perhaps also targets not naturally infected – and the notion that latently infected targets can present antigen stochastically (and moreover directly) to T cells is attractive, given that MHCI expression is likely intact in this setting.
As attractive as this conclusion is, perhaps one should not yet exclude entirely a contribution by infected, professional antigen presenting cells in maintaining the memory T cell pool in chronically infected animals. We know that tissue-derived CD11c+ dendritic cells (DC) are the Trojan Horse for MCMV systemic dissemination during acute infection, and that they defy the DC paradigm by escaping the lymph node to seed peripheral tissues. Whether they continue to recirculate during chronic MCMV infection via draining lymph nodes is not known, nor is their life-span, but their capacity to disseminate broadly may nevertheless contribute to sustaining the pool of tissue-resident memory T cells.
There may be no single mechanism for how MCMV-specific iTEM are maintained, but readouts will be highly dependent on the viruses used (i.e. wild type versus disabled), the host used, and the route of inoculation. For example, previous studies that compare the relative contribution of hemopoietic versus non-hemopoietic cells in iTEM encounters using adoptive transfer of BM to hemo-ablated recipients are complicated by the fact that tissue resident APC are radio-resistant. Studies that use MCMVs inherently deficient in replication – particularly in the salivary glands – are also likely to affect interpretations. While evidence from these studies may help build a case, it is sometimes worthwhile to consider other interpretations. Finally excluding productive infection entirely based on absence of evidence is, (as always) problematic.
Taken together, this data provides unequivocal evidence for active antigen presentation long after resolution of acute infection. I am excited by, but not fully convinced that latently infected cells are the sole contributors to the iTEM reservoir.
Author Response
We thank the reviewer for the valuable comments and the help to enhance the discussion of our manuscript. We improve the discussion regarding the potential role of professional antigen-presenting cells in the induction and maintenance of memory inflation (see lines 232-275).